# Robust microscale structural superlubricity between graphite and nanostructured surface

Xuanyu Huang [1,2,3], Tengfei Li[1,4], Jin Wang [5], Kai Xia[6], Zipei Tan[1,4], Deli Peng[6], Xiaojian Xiang[6], Bin Liu[4], Ming Ma [1,2,3,6] ✉ & Quanshui Zheng [1,2,3,4,6] ✉

Structural superlubricity is a state of nearly zero friction and no wear between two contacted solid surfaces. However, such state has a certain probability of failure due to the edge defects of graphite flake. Here, we achieve robust structural superlubricity state between microscale graphite flakes and nanostructured silicon surfaces under ambient condition. We find that the friction is always less than 1 μN, the differential friction coefficient is on the order of $10^{-4}$, without observable wear. This is attributed to the edge warping of graphite flake on the nanostructured surface under concentrated force, which eliminate the edge interaction between the graphite flake and the substrate. This study not only challenges the traditional understanding in tribology and structural superlubricity that rougher surfaces lead to higher friction and lead to wear, thereby reducing roughness requirements, but also demonstrates that a graphite flake with a single crystal surface that does not come into edge contact with the substrate can consistently achieve robust structural superlubricity state with any non-van der Waals material in atmospheric conditions. Additionally, the study provides a general surface modification method that enables the widespread application of structural superlubricity technology in atmospheric environments.

Friction and wear are two fundamental physical phenomena coupled together in nature, which have caused huge energy waste, environmental pollution and component failure in mechanical system[1], and made it difficult for a large number of key technologies such as novel design of micro-electromechanical system (MEMS)[2]. It is estimated that nearly one-third of the energy provided by fossil fuels in automobiles is dissipated by friction[3]. In the microscopic world, based on the scale effect, the interface friction and wear will become one of the most important problems compared to other effects, which cause the failure of the devices.

Although liquid lubricants such as organic oils have a great inhibitory effect on friction and wear in practical applications, they will fail under strong constraints and extreme external conditions, such as high external loads, high temperatures, as well as in the presence of chemical contamination or in a vacuum environment[4]. Lubrication based on the shear within liquid will fail on nanoscale as the viscosity may greatly increase[5], which is difficult to apply to microscale scenarios, such as MEMS, micro sensors, micro robots, etc[6,7]. Therefore, to solve the above problems, a efficient technology is needed to reduce friction or even eliminate wear from the essential physical characteristics, instead of introducing other indirect substances as friction pairs.

Structural superlubricity (SSL), a state of nearly zero friction and no wear between two contact solid surfaces[6,8,9], that relies on the effective cancellation of lateral forces between rigid crystalline

---

[1]Center for Nano and Micro Mechanics, Tsinghua University, Beijing 100084, China. [2]Department of Mechanical Engineering, Tsinghua University, Beijing 100084, China. [3]State Key Laboratory of Tribology in Advanced Equipment (SKLT), Tsinghua University, Beijing 10084, China. [4]Department of Engineering Mechanics, Tsinghua University, Beijing 100084, China. [5]International School for Advanced Studies, Trieste 34136, Italy. [6]Institute of Superlubricity Technology, Research Institute of Tsinghua University in Shenzhen, Shenzhen 518057, China. ✉e-mail: maming16@tsinghua.edu.cn; zhengqs@tsinghua.edu.cn

contacts provides a disruptive solution to above challenge. The successful achievement of microscale SSL in ambient conditions in 2012 marked a significant milestone[10]. Subsequently, the exploration of high-speed SSL (reaching 293 m/s)[11] has further ignited widespread interest, not only within the realm of academic research but also in practical applications[12]. These applications include superlubric generators (SLGs)[13,14] and superlubric resonators[11,15], showcasing the diverse potential of SSL technology. Notably, recent advancements have expanded the scope of SSL by demonstrating its effectiveness between microscale graphite flakes and various non-van der Waals materials, such as diamond-like-carbon (DLC), silicon (Si), silicon dioxide ($SiO_2$), aluminum oxide ($Al_3O_2$), silicon nitride ($Si_3N_4$), and others[16]. This major development significantly broadens the possibilities for implementing SSL in a range of applications.

However, in this study, we unexpectedly discovered that when an atomic force microscope probe is pressed on a microscale graphite flake and dragged to slide on an atomic-level flat silicon surface, there is a certain probability of high friction (>5 µN) and corresponding wear will occur. On the contrary, the friction is always less than 1 µN between graphite flake and nanostructured silicon surface, the differential friction coefficient is on the order of $10^{-4}$, and there is no observable wear, indicating the presence of SSL. Through detailed characterization and simulation, we found that friction and wear between the graphite flake and atomic-level flat silicon surface mainly occur along the edge, and the dangling bonds along the edge will have a strong interaction with the substrate atoms[17–19]; but the edge of graphite flake on the nanostructured surface will occur warping under concentrated force, thereby reducing the edge interaction between the graphite flake and the substrate.

## Results

### Ultralow friction between graphite flake and nanostructured surfaces

To commence, we initiated an assessment of the friction force between micron-sized graphite flakes and nanostructured silicon, comparing it with the friction force encountered at the graphite/flat silicon interface. The preparation of the graphite/silicon interface involved a two-fold procedure[14]. In the initial step, our attention was directed the selection of a graphite flake that underwent shear-induced cleavage from a 4 × 4 × 0.9 µm square graphite mesa, leveraging its inherent self-retracting motion (SRM) property[20] (refer to the Methods section and Supplementary Section 3 for comprehensive details). The graphite mesa itself fabricated of highly ordered pyrolytic graphite (HOPG) and was adorned with a 100 nm thick layer of gold (fabrication specifics can be found in Supplementary Section 1 and the Methods section). The SRM motion is a strong indication for the presence of SSL at the interface between the bottom of graphite flake and the surface of the lower part of the graphite mesa. It is noteworthy that the cleaved surface, often referred to as the bottom surface, exhibited the characteristics of a single-crystalline graphene sheet, further ensuring a pristine and homogenous foundation for subsequent measurements[21].

In the subsequent step, the graphite flake was delicately transferred onto two distinct types of silicon surfaces, as expounded upon in the Methods section and Supplementary Section 3 for comprehensive elucidation. One of them is a very clean and atomic-level flat silicon surface, and the other is a surface with nanostructures, their surface topography is shown in Fig. 1c and e, respectively. The scanning electron microscopy (SEM) observations of graphite/nanostructured silicon interface are shown in Supplementary Fig. 5. The selection and transfer process of graphite flake were performed under an ambient atmospheric environment in a thousand-class clean room with a temperature of 25 ± 1 °C and a relative humidity of 25 ± 1%.

After completing the above steps, we used the lateral force module of the atomic force microscope (AFM, Cypher S-Oxford Instruments) to measure the friction between the graphite flake and

two types of silicon surfaces, and the experimental setup is shown in Fig. 1a, which includes the AFM tip, laser, optical lens, and PZT tube (see the Methods section and Supplementary Section 5 for details). We applied pressure on the upper surface of the graphite flake utilizing an AFM tip, aiming to induce controlled sliding between the graphite flake and the two distinct silicon surfaces. This sliding motion was achieved by employing the lateral movement capabilities of the piezoelectric displacement platform. Figure 1b presents an optical image capturing the experimental setup. The laser will be irradiated on the cantilever of the AFM tip, and the photodiode will receive the reflection laser signal to feel the deformation of the cantilever (See Methods section and Supplementary Section 5 for more details). The friction measurements were performed under an ambient atmosphere with a temperature of 25 ± 1 °C and a relative humidity of 30 ± 4%.

The measured lateral force signals when the graphite flake slides on the atomic-level smooth surface (Fig. 1c) and the nanostructured surface (Fig. 1e) are shown in Fig. 1d and f, respectively. For a normal load of 20.04 µN, we found that the friction of graphite flake sliding on the atomic-level smooth surface is ~6 µN, which is much larger than that sliding on the nanostructured surface, which is ~0.2 µN. This high friction (~6 µN) is also orders larger than the typical value for graphite flake sliding on other surfaces showing SSL, which is on the order of 0.1 µN[16]. To understand the phenomenon of the large friction between the graphite flake and the atomic-level flat silicon surface in Fig. 1d, we performed detailed in situ characterizations of the sliding interface as shown in Supplementary Section 7. We observed that wear occurs at the boundary of the sliding region, and we speculate that the possible reason that the carbon atoms at the edge of the graphite flake are dragged by the substrate through the strong chemical interaction with silicon, resulting in the occurrence of large friction and wear (i.e., failure of SSL state).

### Robust SSL state at the graphite flake/nanostructured silicon interface

To further verify the robust structural superlubric (SSL) state between the graphite flake and the nanostructured silicon surface, we employed polystyrene microspheres as a mask to fabricate nanostructures with uniform peak heights and distribution on the silicon surface (the fabrication process is presented in the Supplementary Section 2 and Methods section). Subsequently, a series of tribological tests were conducted using the same experimental setup as shown in Fig. 1a and under identical environmental conditions. These tests involved examining the interaction between a 4 × 4 µm graphite flake and a silicon surface with specially prepared nanostructures. The detailed testing procedures can be found in the Methods section and Supplementary Section 6. Firstly, we carefully selected a silicon area featuring nanostructures, as illustrated in Fig. 2b. Subsequently, we conducted in situ measurements of the friction force at the graphite flake/nanostructured silicon interface under different normal loads, as depicted in Fig. 2d. Remarkably, we observed an extraordinarily low differential friction coefficient of 0.00044 ± 0.00027, with the friction force measuring approximately 0.7 µN. Secondly, we performed a sliding experiment consisting of 5120 cycles on the same area using the graphite flake. The friction forces encountered throughout the entire sliding process are displayed in Fig. 2a. Interestingly, the friction forces exhibited a stable range of 0.6–0.7 µN. Following the sliding experiment, we conducted an in situ characterization of the nanostructured silicon surface's morphology, as shown in Fig. 2c. No discernible wear debris or damage was observed.

Further, we performed Raman mapping characterization on the slid nanostructured silicon surface, the red frame in Fig. 2e is the scan area, which includes the whole sliding region (yellow dash frame). The intensity distributions of $D$ peak (1350 $cm^{-1}$), $G$ peak (1580 $cm^{-1}$) and $2D$ peak (2700 $cm^{-1}$) are displayed in Fig. 2f–h, respectively, where Fig. 2i shows the single-point Raman spectrum at the blue cross mark position

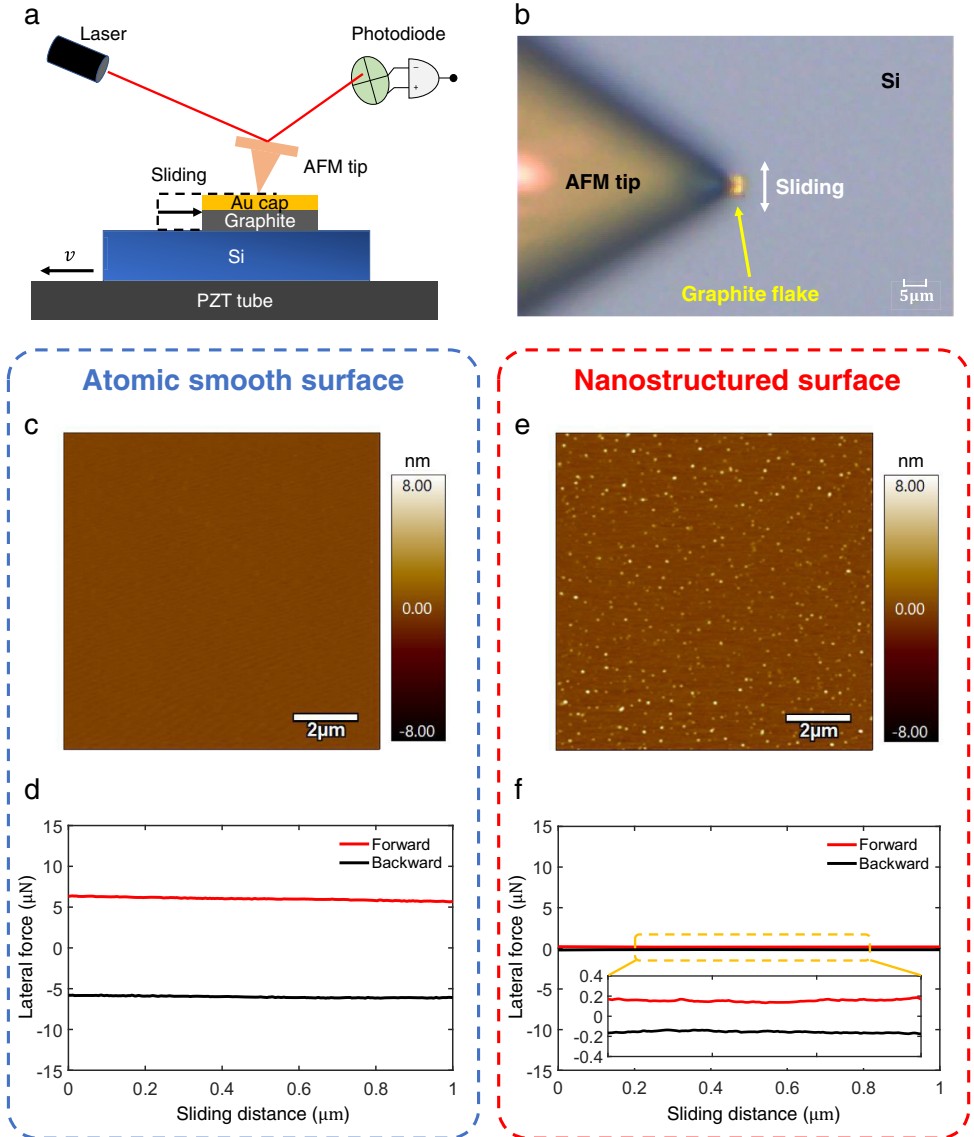

**Fig. 1 | Friction measurement of superlubric graphite flake and two types of silicon surfaces. a** Schematic diagram of the experimental setup to measure the friction of graphite/silicon interface. The silicon substrate was rigidly fixed to the piezoelectric ceramic transducer (PZT) stage on top of which an Au capped graphite flake was placed. Normal force and lateral force are applied by pressing the AFM tip on the top of the Au cap. When the PZT stage reciprocates with velocity $v$, the AFM tip will drag the graphite flake to relative slide on the silicon substrate.

**b** Optical microscopic image of experimental set-up. **c** and **e** are the surface topography of atomic-level smooth surface and nanostructured surface, respectively. **d**, **f** The lateral force signal when graphite flake sliding on the atomic-level smooth surface and nanostructured surface respectively, with a displacement amplitude of 1 μm and a speed of 2 μm/s under a normal force of 20.04 μN. Source data are provided as a Source Data file.

in Fig. 2e (illustration is an enlarged view of range 1200–2800 cm$^{-1}$). No observable $D$, $G$, $2D$ graphite peaks were found, indicates that there was no graphite wear debris on the slid nanostructured silicon surface. We also performed Raman characterization at different positions (points 1–9 in illustration) on the slid graphite flake surface, as shown in Fig. 2j. The absence of the $D$ peak indicates the absence of visible wear on the surface of the graphite flake after undergoing 5120 sliding cycles. Altogether, the small friction (<1 μN), ultralow differential friction coefficient (~10$^{-4}$), and absence of wear indicate the presence of robust SSL state of graphite/nanostructured silicon interface.

### Mechanism of robust SSL state at graphite flake/nanostructured silicon interface

To understand the mechanism of the SSL state at graphite/nanostructured silicon interface, we carried out simulations using finite element method (FEM) for the experimental process as shown in Fig. 3a. The van der Waals interaction between the graphite flake and silicon surface is derived from Lennard–Jones (LJ) potential (see Supplementary Section 8.1 for details). The graphite flakes are 4 μm × 4 μm in size and 200 nm in height, covered with 100 nm thick Au caps and the normal load on its center is 20 μN, which is within the range of the applied normal load in the experiments. For the nanostructured silicon surface, there has been research work that enables accurate meshing of rough surface while keeping the real morphology of the surface by the method of self-fractal surfaces, which can accurately simulate the contact state in real situations[22,23]. However, for a clearer and more concise analysis of the mechanism, we simplified the nanostructured surface as a plain surface with array of regular and gentle undulations. The spacing and height of the rough peak array are 400 nm and 7 nm respectively, and the surface shape of each roughness peak is obtained

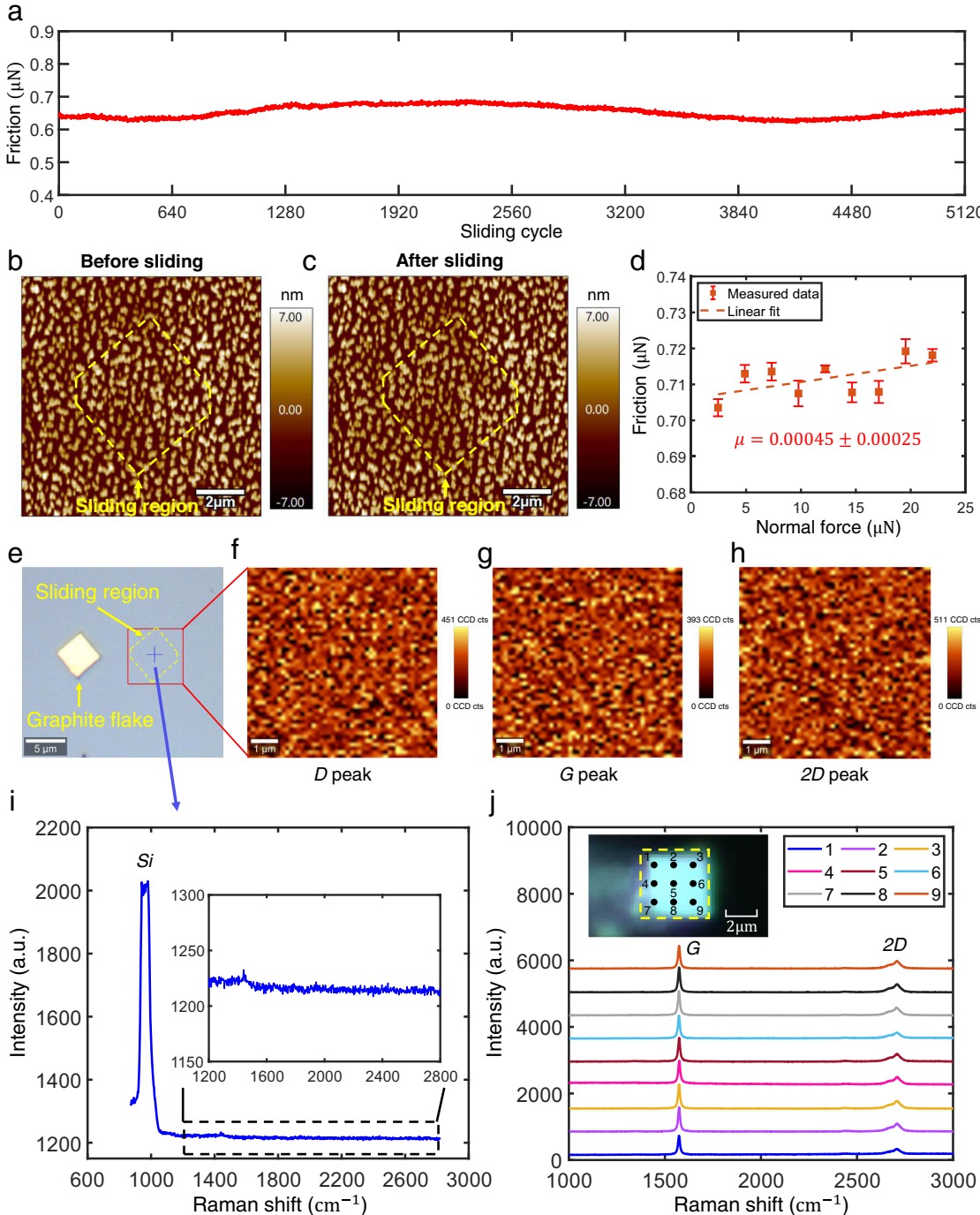

**Fig. 2 | Tribological tests between graphite flakes and nanostructured silicon surface. a** Measured friction force during 5120 continuous sliding cycles with a displacement amplitude of 1 μm and speed of 2 μm/s under a normal force of 14.65 μN. **b** and **c** are the topography of the nanostructured surface before and after 5120 sliding cycles, respectively, where the yellow dashed frame is the boundary of the sliding region. **d** Measured friction force of graphite flake/nanostructured silicon interface in situ under different normal load with a displacement amplitude of 1 μm and speed of 2 μm/s; each point and error bar were obtained by 10 tests and had friction coefficients of 0.00045 ± 0.00025. **e–h** The Raman characterization results of the sliding region on nanostructured silicon, where **e** is the optical observation, the red frame is the scan area of 6 μm × 6 μm (including the whole sliding region), **f–h** The intensity distributions of *D* peak (1350 cm⁻¹), *G* peak (1580 cm⁻¹) and *2D* peak (2700 cm⁻¹), respectively, there is no observable graphite composition. **i** Single-point Raman spectrum at the blue cross mark position in **e**, the illustration is an enlarged view of the black dash frame region, there are no observable *D*, *G* and *2D* graphite peaks. **j** Raman characterization of the bottom surface of graphite flake after 5120 sliding cycles; points 1–9 represent the test positions, there is no observable *D* peak, and the illustration is the optical observation of flipped graphite flake. Source data are provided as a Source Data file.

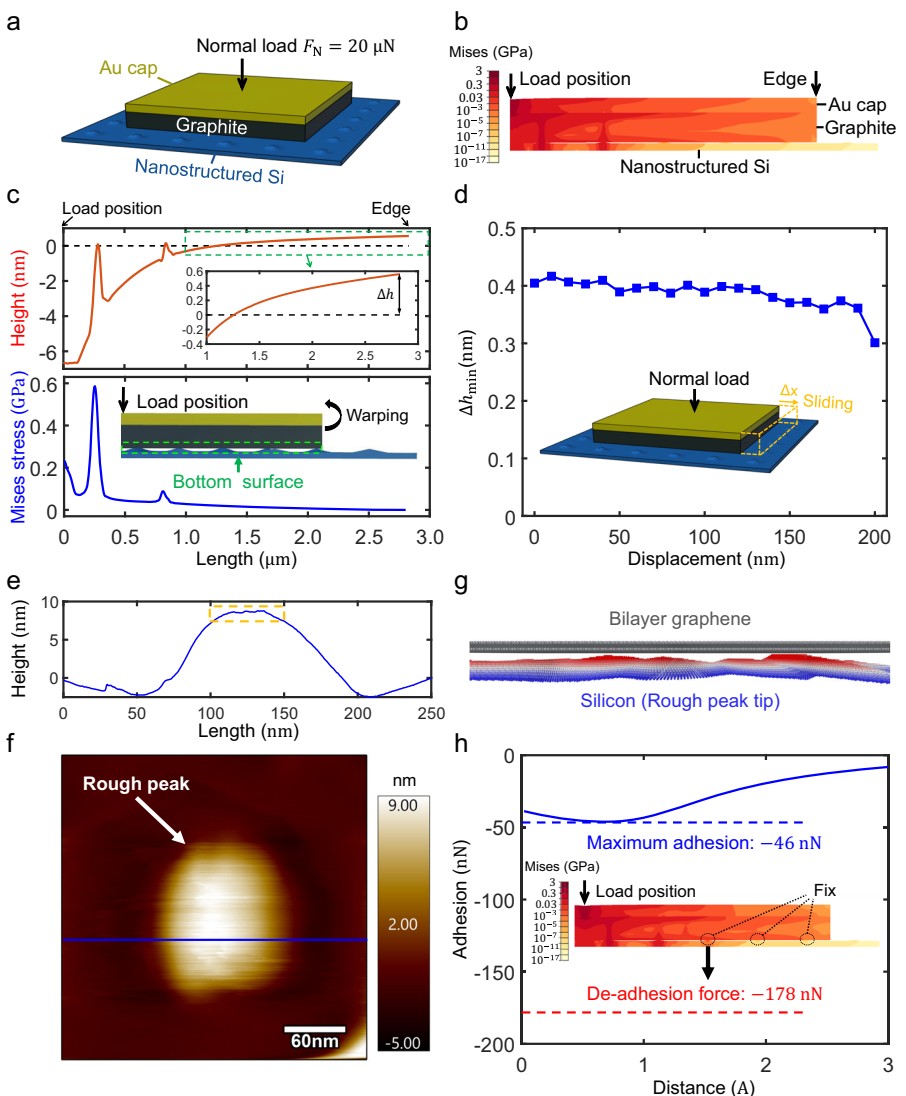

**Fig. 3 | Edge warping of graphite flakes on nanostructured surface. a–d** The finite element simulation results. **a** Schematic diagram of the finite element simulation model with nanostructured silicon/graphite flake heterojunction, where the concentrated normal load of 20 μN is applied to the center of the graphite flake. **b** Mises stress distribution across the diagonal section under central normal loading when $\Delta x = 0$. **c** Displacement in height (upper) and Mises stress (lower) distribution along the bottom surface of graphite flake across diagonal section (see illustration in the lower image), where the inset in the upper image is a partial enlarged view of the green dashed frame, and the black dotted line in the upper image is the original position of the bottom surface of the graphite flake, the warping height of edge is $\Delta h$. **d** Minimum warpage height along the edge $\Delta h_{min}$ under different displacement $\Delta x$ of graphite flake (simulation range is half the period of rough peak of 200 nm, and the dot pitch is 10 nm). **e–h** The molecular simulation results. **e** and **f** are the morphology of a typical single rough peak, where

(**f**) is the 250 nm × 250 nm two-dimensional morphology, and **e** is the height data of the blue section line in (**f**). **g** The molecular dynamic simulation model of graphite/single-coarse peak interface, the rough peak tip in the model corresponds to the part in the orange dashed box in (**e**), and the simulation box size is 60 nm × 60 nm × 7 nm. **h** Simulated adhesive force upon with the distance respect to the equilibrium interlayer spacing (blue solid line), where the maximum adhesion force is 46 nN (blue dotted line). The illustration is the Mises stress distribution across the diagonal section under central normal loading by constraining the warpage of the bottom surface of the graphite flake, i.e., the relative displacement in the positive z direction with the outer rough peak of the nanostructured silicon, where the force needed to balance the bending moment around the third rough peak (i.e., de-adhesion force) is about 178 nN (red dotted line), which is larger than maximum adhesion force. Source data are provided as a Source Data file.

by rotating and sweeping the function $y = A\sin\left(\frac{\pi x}{L}\right)^n$ around the symmetry axis, where $A = 7$ nm, $L = 350$ nm and $n = 10$. The above parameters are statistically obtained from actual measured topography from Fig. 2b (see Supplementary Section 4 for details). The sliding displacement $\Delta x$ of the graphite flake is 0 in the initial state.

When the graphite flake is pressed against nanostructured silicon surface, as shown in Fig. 3b for the Mises stress distribution across the diagonal section, there is a concentrated normal stress around the rough peak near the loading center, while no normal stress between the peripheral region of the graphite flake and the nanostructured

silicon is observed. We further plot the displacement in height (upper) and Mises stress (lower) distribution along the bottom of the graphite flake pressed on nanostructured silicon across the diagonal section, as shown in Fig. 3c, which shows that the peripheral part of the graphite flake warps upon the original state (black dotted line) under the central force, where the warping height of the edge is $\Delta h$. Together with the zero normal stress between the peripheral region of the graphite flake and the nanostructured silicon, it is evident that the graphite edges become out of contact upon loading (Supplementary Section 8.4 gives the overall pressure distribution between the bottom surface of the

graphite flake and the nanostructured silicon substrate, indicating that only the rough peaks near the loading center are in contact with the bottom surface of the graphite flake, and the corresponding actual contact area is only $6.23 \times 10^{-2} \mu m^2$). Such transition of the contact state, however, is absent for graphite flake pressed against a flat silicon surface (see Supplementary Section 8.3 for details). We then simulated the edge warpage of graphite flake during sliding on the nanostructured silicon surface through half rough peak period (i.e., 200 nm), and plotted the relationship between the minimum warpage height along the edge $\Delta h_{min}$ and sliding distance $\Delta x$ in Fig. 3d. The minimum warpage height $\Delta h_{min}$ goes lower when the load point goes closer to the rough peak, but remains positive throughout, which indicates that the warping of the edge keeps stable during the continuous sliding. From mechanical point of view, the FEM simulation shows that the bending moment generated by the central load on the nanostructured silicon surface overcomes van der Waals adsorption between graphite and nanostructured silicon surfaces and results in the warping of the graphite flake edges during sliding. In addition, the maximum stress at the bottom surface of graphite flake shown in the lower image of Fig. 3c is around 0.6 GPa, which is lower than the critical withstand pressure to maintain the SSL state at the bottom surface of graphite flake (>1 GPa)[24], so stress concentration will not lead to the failure of the SSL state within the range of normal force applied in our experiments.

We also performed finite element method (FEM) simulations of the warping of graphite flakes that placed on nanostructured silicon with different separation $d$ under central loading, as detailed in Supplementary Section 8.5. From the results, we can conclude that the warpage height of the graphite flake edge decreases with the increase of the separation between the nanorough peaks (Supplementary Figs. 16c–e). When the nanorough peaks becomes sparse enough ($d$>600$nm$), even increasing the loading at the center of the graphite flake cannot warp the entire edge above the horizontal plane (Supplementary Fig. 16c and e). As a result, the part of the edge that sinks below the horizontal plane will interact with the nanorough peaks of the substrate (collisions, bonding, breaking, etc.), regenerating high friction and wear, thereby disrupting the robust SSL state. Therefore, controlling the size and separation of the nano-asperities is of great significance for the surface modification method in this work to achieve robust SSL, and the experimental research to verify the above-mentioned finite element simulation prediction law is the work that needs to be carried out in the next stage.

Considering that finite element methods in a relatively large scale may underestimate or overestimate the van der Waals interaction between contact surfaces which acts at subnanometer scale, we performed additional molecular dynamics (MD) simulations to extract the maximum adhesion between one rough peak and the graphite. The MD simulation model was established according to the morphology of the rough peak in experiments (Fig. 3e, f), is shown in Fig. 3g where the simulation box has the size of 60 nm × 60 nm × 7 nm (see Methods section for details). The simulated adhesive force versus distance respect to the equilibrium interlayer spacing is shown in Fig. 3h (blue solid line), where the maximum adhesion is 46 nN (blue dotted line). To quantify the de-adhesive force created by the bending moment of the graphite flake on the rough peak of a nanostructured silicon surface under concentrated force, we modified the boundary conditions of the finite element method (FEM) simulation depicted in Fig. 3a. By constraining the warpage of the bottom surface of the graphite flake, i.e., the relative displacement in the positive z direction with the outer rough peak of the nanostructured silicon, we calculated the Mises stress distribution across the diagonal section under central normal loading, as shown in the inset of Fig. 3h. Our simulation results revealed that along the diagonal direction, the two closest rough peaks to the loading center experience pressure, while the third rough peak

generates the balancing reaction force (i.e., the de-adhesive force), which was found to be approximately 178 nN (represented by the red dotted line in Fig. 3h), surpassing the maximum adhesion force of 46 nN. As a result, de-adhesion occurs at the third rough peak position. Moreover, the third rough peak closest to the loading center experiences the lowest de-adhesive force, so subsequent de-adhesion will occur at all peripheral rough peaks after de-adhesion occurs at the third rough peak. These MD and FEM results support the de-adhesion process at the atomic scale.

Based on the above analysis, we can conclude that the mechanism of the robust SSL state at the graphite flake/nanostructure silicon interface is that the edge of the graphite flake on the nanostructure silicon surface will warp under the central normal force, thereby eliminate the interaction between the edge of graphite flake and substrate, which is proved to be the main source of friction[18]. On the contrary, due to the tight van der Waal adhesion between the graphite flake and atomic flat silicon surface, the edges of the graphite flakes interact with the silicon surface, which may eventually lead to large friction and induce wear.

In addition, through contact angle measurements, we found that water exhibited weak hydrophobicity on the surface of the prepared nanostructured silicon (Supplementary Fig. 18 and Supplementary Table 3), so it can be judged that in our experimental environment, that is, the temperature of $25 \pm 1 °C$ and the relative humidity of $30 \pm 4\%$, the water film adsorbed on the prepared nanostructured surface is very thin (<1 nm[25]) that it does not completely submerge the nanostructures, and the water film covering the apex of the nanostructure will be extruded under the pressure in our experimental conditions (detailed analysis see Supplementary Section 10).

## Experimental verification of edge warping mechanism

We verified above proposed mechanism by conducting a series of experiments for graphite flake/nanostructured silicon interface. First of all, we measured the friction coefficient of graphite flake/nanostructured silicon interface for different loading positions, where the eccentric distance, i.e., distance between the loading position and the center of the flake is $\delta$, as shown in Fig. 4a, where the optical observation of the different $\delta = 0, 0.4, 0.8, 1.2 \mu m$ are shown in Fig. 4b. The measured friction force of different applied normal force under different $\delta$ are shown in Fig. 4f, which can be seen that the friction force and fitted friction coefficient increases rapidly with the increase of $\delta$, especially when $\delta = 1.2 \mu m$, that is, when AFM tip pressed near the edge of graphite flake.

We then simulated the warping of graphite flake on nanostructured silicon under different $\delta$ through the same FEM method in the last section. The simulated height distribution of the graphite flake bottom surface under a normal loading of 20 μN with $\delta = 0$ and $\delta = 1.2 \mu m$ are shown in Fig. 4c, d, respectively. Under central normal loading ($\delta = 0$) condition, all edges of the graphite flake are de-adhered and warped due to bending moment ($\Delta h_{min}$>0). On the contrary, when the edges are loaded ($\delta = 1.2 \mu m$), the edges near the loaded side ($X = 2 \mu m$) cannot be warped due to insufficient bending moments and concentrated forces, resulting in downwards deformation ($\Delta h_{min}$<0). The minimum warpage height $\Delta h_{min}$ under different loading positions ($\delta = 0, 0.4, 0.8, 1.2, 1.6 \mu m$) is shown in Fig. 4e, where $\Delta h_{min}$ drops rapidly to negative when $\delta \geq 1.2 \mu m$, which explains the failure of the SSL state when $\delta = 1.2 \mu m$ in the experiment (high friction force and coefficient in Fig. 4f when $\delta = 1.2 \mu m$). Therefore, above results indicates that the incretion of the interaction between the graphite flake edge and the substrate increases the friction force and friction coefficient (SSL state failure)[18], which verify the edge warping under concentrated normal force in the center of the graphite flake is the mechanism of robust SSL state at graphite flake/nanostructured silicon interface.

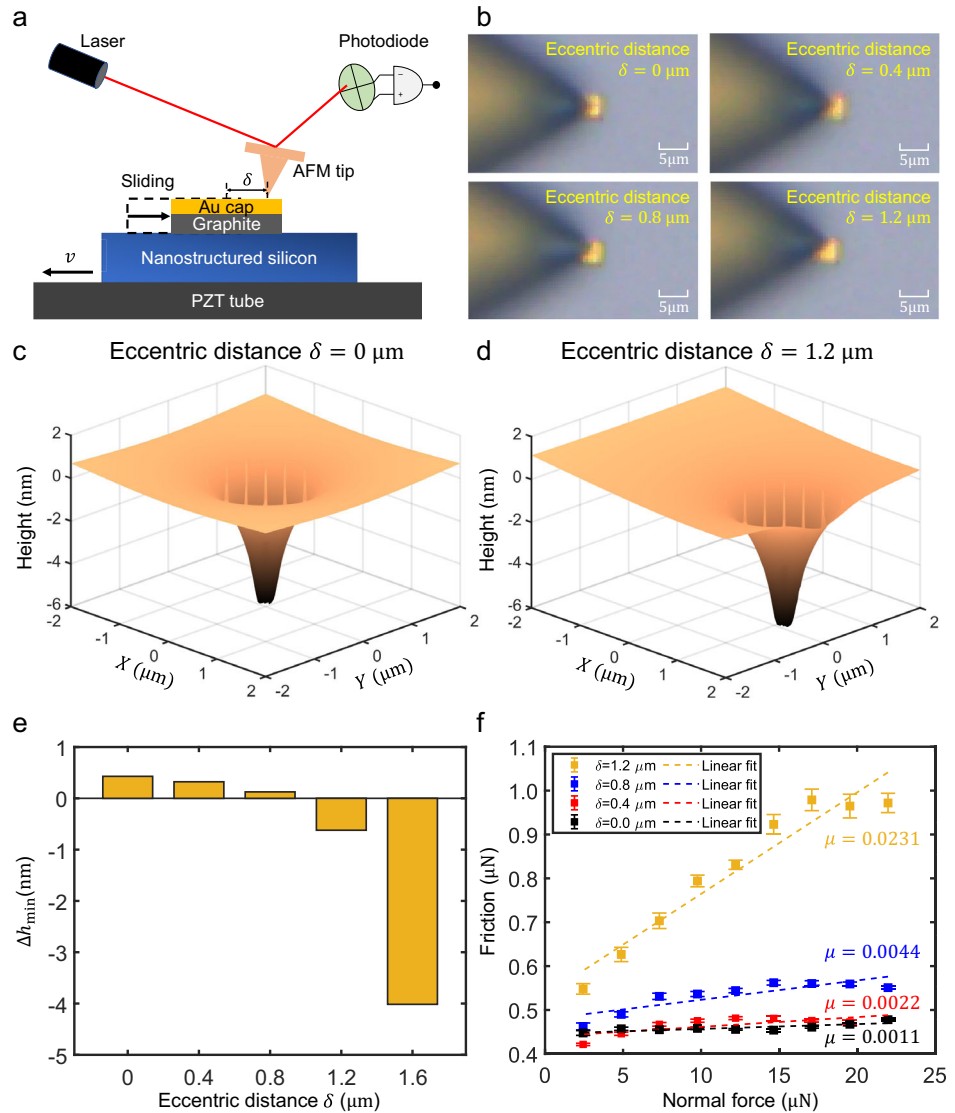

**Fig. 4 | Friction coefficient between graphite flake and nanostructured surface for different loading positions. a** Schematic diagram of the experimental set-up to measure the friction of graphite flake/nanostructured silicon interface under different loading positions, where the distance between the loading position of the AFM tip and the center (eccentric distance) is $\delta$. **b** Optical microscope observation for different $\delta$. **c**, **d** Simulated height distribution of the graphite flake bottom surface under the normal loading when $\delta = 0$ and $\delta = 1.2\,\mu m$ respectively with a normal load of 20μN. **e** Relationship between minimum warpage height along the edge $\Delta h_{min}$ and eccentric distance $\delta$ with a normal load of 20μN. **f** Measured friction force of different applied normal load under different $\delta$; each point and error bar were obtained by 15 tests, and the fitted friction coefficients of $\delta = 0, 0.4, 0.8, 1.2\,\mu m$ are 0.0011, 0.0022, 0.0044 and 0.0231, respectively. Source data are provided as a Source Data file.

An intuitive deduction according to the mechanism as well as Fig. 4 is larger eccentricity will facilitate the failure of SSL. To this end, we conducted a series of tribological tests for $\delta = 1.8\,\mu m$. As shown in Fig. 5a, the measured friction fluctuated and suddenly increased during the process of 3072-cycle sliding, which is probably caused by the collision between the edge of the graphite flake and the nanostructure. Meanwhile, wear debris is also observed on nanostructured silicon surface after sliding (Fig. 5b, c). Further Raman spectroscopy map of the nanostructured silicon surface after sliding (Fig. 5d–h) shows that the wear debris is composed of graphite (contain *D*, *G*, *2D* graphite peaks). This is also supported by the direct Raman spectroscopy characterization for the bottom surface of graphite flake after sliding (Fig. 5i), where the *D* peak representing defects appears at the edge of the slid graphite flake surface perpendicular to the sliding direction (points 1–3 in illustration of Fig. 5i).

## Discussion

In traditional tribology and structural superlubricity theories, it is generally believed that rougher surfaces lead to higher friction and lead to wear. However, this study found that the graphite flake/flat silicon interface has a tendency to exhibit high friction (>5 μN) and wear. On the other hand, the microscale graphite flake/nanostructured silicon interface consistently displays a robust structural superlubric (SSL) state under ambient conditions, characterized by friction force less than 1 μN, a friction coefficient of approximately $10^{-4}$, and no observable wear. The mechanism behind this phenomenon is revealed to be edge warping of the graphite flake on the nanostructured surface under concentrated force, which effectively eliminates high friction and wear resulting from strong edge-substrate interactions[17,18]. This work not only challenges traditional tribology and structural superlubricity theories which usually predicts a larger friction for rougher

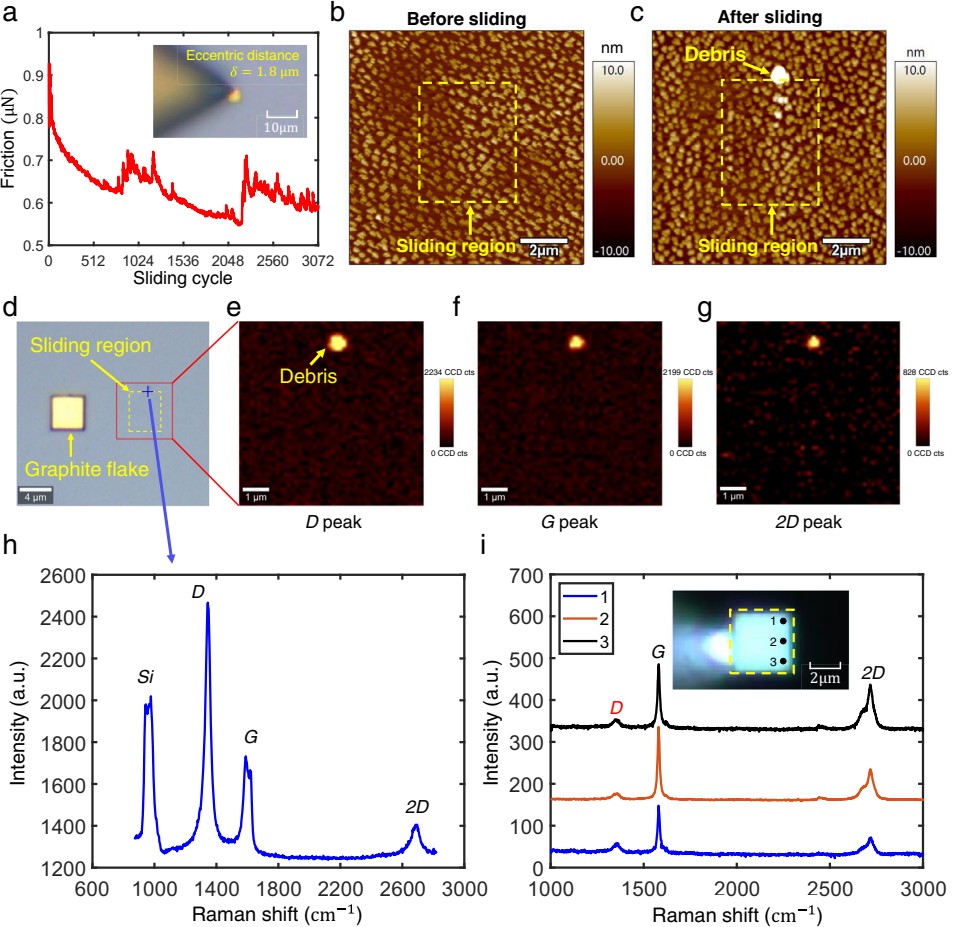

**Fig. 5 | Wear characterization of the graphite flake on the nanostructured surface under edge loading. a** Measured friction force during 3072 continuous sliding cycles with a displacement amplitude of 1 μm and speed of 2 μm/s under a normal force of 21.98 μN. **b** and **c** are the topography of the nanostructured surface before and after 3072 sliding cycles, respectively, where the newly appearing highlighted region is the wear debris, and the yellow dashed frame is the boundary of the sliding region. **d–h** The Raman characterization results of the sliding region on nanostructured silicon, where **d** is the optical observation, the red frame is the scan area of 6 μm × 6 μm (including the whole sliding region), **e–g** are the intensity distributions of $D$ peak (1350 cm$^{-1}$), $G$ peak (1580 cm$^{-1}$) and $2D$ peak (2700 cm$^{-1}$), respectively, it can be seen that there is large intensity at the location of the wear debris in three images. **h** Single-point Raman spectrum at the wear debris position (blue cross mark in **d**), there are significant $D$, $G$ and $2D$ graphite peaks. **i** Raman characterization of the graphite flake bottom surface at edge positions on the side close to the loading position after 3072 sliding cycles; points 1–3 represent the test positions, where significant $D$ peaks appear. Source data are provided as a Source Data file.

surface, reduces the roughness requirements for SSL technology which used be a challenge for practical application of SSL, but also demonstrate that a graphite flake with a single crystal surface that does not come into edge contact with the substrate can consistently achieve robust SSL state with any non-van der Waals material in atmospheric conditions. Furthermore, the study provides a general surface modification method to achieve robust SSL state between graphene flakes and non-vdW materials, further promoting the widespread application of SSL technology. For example, superlubric generators (SLGs) is designed to produce stable high current density and long-lifespan output under relative sliding through SSL technology, and the conversion efficiency is close to 100%[13,14]. However, the edges of graphite flake will cause high friction and wear with a certain probability, leading to the failure of the SLG. Whereas with the surface modification and center-loaded connection of this work, a robust SSL contact can be achieved, avoiding the high friction and wear that can occur at the edges of graphite flakes (refer to Supplementary Section 11 for more information), promoting SLG to large-scale batch applications. While the microfabrication process of such structure is under operation and takes time, we believe that with the mechanism revealed in

our present manuscript, it is reasonable to believe that the friction and wear will greatly be reduced with the help of edge warping.

## Methods
### Fabrication of graphite mesa with Au cap[14]
We fabricated square arrays of graphite mesas with an Au cap on highly ordered pyrolytic graphite (HOPG), specifically of ZYB grade (Brucker)[26]. The detailed fabrication process is depicted in Supplementary Fig. 1a. Initially, a double layer of photoresist consisting of LOR 1 A (100 nm) and ZEP (400 nm) was spun onto the freshly cleaved surface of the HOPG, as illustrated in Supplementary Fig. 1a (i). Next, the photoresist in the region corresponding to the mesa pattern array was selectively removed through an electron beam lithography and development process, as depicted in Supplementary Fig. 1a (ii). Subsequently, an Au film with a thickness of 100 nm was deposited onto the surface via electron beam evaporation, as shown in Supplementary Fig. 1a (iii). By employing a lift-off process, the Au pattern array was obtained, as illustrated in Supplementary Fig. 1a (iv). Finally, utilizing the Au pattern array as a mask, we achieved the formation of graphite mesas with an Au cap through an oxygen reactive ion etching process,

as shown in Supplementary Fig. 1a (v). The etching depth was measured to be 0.8 μm. The characterizations of the fabricated graphite mesas with the Au film are presented in Supplementary Fig. 1b–e.

## Fabrication of nanostructured silicon surface

Firstly, we ultrasonically cleaned the silicon surface with acetone, alcohol, and deionized water for 5 min, and then performed the surface hydrophilic treatment by oxygen plasma, as shown in Supplementary Fig. 2a (i). Then we mixed 200 nm polystyrene (PS) microsphere solution (mass fraction 10%, produced by Huge Biotechnology Company) with alcohol and deionized water in a ratio of 1:2:2 and ultrasonic for 30 min to obtain the diluted solution. Further we immersed the silicon surface with deionized water and used the pipette to drop the diluted solution 250 μL, and made it self-assemble on the liquid interface, as shown in Supplementary Fig. 2a (ii). Secondly, we evaporated the deionized water in atmospheric environment, and deposit the self-assembled PS microsphere array on the silicon surface, as shown in Supplementary Fig. 2a (iii). Then, we reduced the size of PS microspheres by oxygen plasma etching (power: 400 W, oxygen flow: 400 sccm, etching time: 4 min), as shown in Supplementary Fig. 2a (iv). Thirdly, we used PS microsphere array as mask to obtain the nanostructures by ion beam etching process (beam: $1 \text{ mA/cm}^2$, energy: 500 eV, etching time: 15 s), as shown in Supplementary Fig. 2a (v). Lastly, we removed the PS microsphere array by excessive oxygen plasma etching, as shown in Supplementary Fig. 2a (vi). Then we use acetone, alcohol, and deionized water ultrasonic cleaning to remove the residual solvents and polystyrene spheres on the nanostructures, and the AFM morphology of the fabricated nanostructured after final ultrasonic cleaning is shown in Supplementary Fig. 2g. The nanostructures are uniformly distributed, and the peak height is uniform. Furthermore, we performed the X-ray photoelectron spectroscopy (XPS, Thermo Fisher, Model: ESCALAB 250Xi, Source gun type: Al K Alpha, Spot size: 500 μm, Lens mode: Standard, Pass energy 30.0 eV) to characterize the prepared nanostructured silicon surface to demonstrate that solvents and polystyrene spheres were completely expelled after the fabrication of the nanostructured surface, as shown in Supplementary Fig. 3.

## The transfer process of graphite flakes to the silicon surface[14]

Firstly, we utilized a tungsten microtip, controlled by a micromanipulator (Kleindiek MM3A), to attach the Au cap of the graphite mesa fabricated as described in Supplementary Fig. 1. This process is illustrated in Supplementary Fig. 4a. By applying shear stress using the micromanipulator, we achieved a controlled splitting of the graphite mesa at approximately 2 μm from the vertical direction, as shown in Supplementary Fig. 4b. Next, we removed the microtip to observe whether the sheared graphite flake underwent self-recovery motion (SRM)[20], which would indicate the presence of a single crystal structural superlubric bottom surface[21]. This observation is depicted in Supplementary Fig. 4c. Subsequently, we re-attached the graphite flake, which exhibited SRM properties, using the microtip and completed the splitting process to completely separate the flake. We then picked up the dangling graphite flake, which was being dragged by the microtip, as shown in Supplementary Figs. 4d, e. Finally, we carefully placed the dangling graphite flake onto the fabricated silicon surface using the micromanipulator. Due to the stronger adsorption force between the graphite flake and silicon compared to that between the microtip and graphite flake, the graphite flake remained on the silicon surface, as depicted in Supplementary Fig. 4f. This process resulted in the formation of the graphite flake/silicon interface shown in Fig. 1a.

## Friction measurement based on AFM system[14]

Friction measurements of the graphite/n-Si heterostructures were conducted in an ambient atmosphere at a temperature of $25 \pm 1\,^{\circ}\text{C}$ and a relative humidity of $30 \pm 4\%$. The experimental setup consisted of a commercial NTEGRA upright AFM (Cypher S-Oxford Instruments), a XYZ piezoelectric displacement platform, a high numerical aperture objective lens (×20), and a visualized AFM tip (ACCESS-NC-GG (Appnano)). A schematic of the experimental setup is presented in Fig. 1a. To ensure precision, we carefully applied pressure to the Au cap of the graphite flake using the AFM tip, employing both the optical microscope and the piezoelectric displacement platform. The in situ calibration of the AFM tip was performed using the Sader method[27,28] for measuring normal direction forces and the diamagnetic levitation spring system[29] for measuring lateral direction forces. Specific results of the calibration process are provided in Supplementary Fig. 8. The silicon's bottom surface is securely affixed to the piezoelectric stage using tape. A photodiode captures the reflection of a laser beam projected onto the cantilever of the AFM tip, enabling precise measurement of lateral force during the sliding process by tracking the movement of the laser spot resulting from cantilever deformation.

## Interface characterization method for tribological experiment[14]

Here, we provide a comprehensive description of the experimental process and methods, as illustrated in Fig. 2, Fig. 5, and Supplementary Fig. 10. For detailed information, please refer to Supplementary Section 6. As an example, we will focus on Fig. 5 to outline the key steps involved. The first step involves utilizing an atomic force microscope (AFM, Cypher S-Oxford Instruments) in tapping mode to characterize the morphology of the silicon surface prior to sliding (Fig. 5b). In the second step, the AFM tip (ACCESS-NC-GG (Appnano)) is used to apply a normal load to the graphite flake and move it to the center of the previously characterized region, as depicted in Supplementary Fig. 9c. The friction force of the sliding process, consisting of 3,072 cycles, is then measured (Fig. 5a). Moving on to the third step, the AFM tip is employed to relocate the graphite flake out of the sliding region, as shown in Supplementary Fig. 9e. Subsequently, the in situ positioning function of the AFM is utilized in tapping mode to characterize the morphology of the sliding region, allowing for the assessment of any observable damage (Fig. 5c). For the fourth step, a Raman characterization (WITec Raman spectra, Ulm, Germany equipped with an alpha 300RA microscope using ZEISS 100x/0.9 objective, a 532 nm laser, a 300 gr/mm grating and a charge-coupled device cooled down to -60°C. The laser power was set 6 mW and the Raman image of the sample was measured with a 200 nm step size and an acquisition time of 1-5 s. The acquired images were analyzed using a WITec Control/Project 5.3.) is performed on the slid silicon surface (Fig. 5d–h) to determine the presence of any observable graphite wear debris based on the presence of $D$ ($1350 \text{ cm}^{-1}$), $G$ ($1580 \text{ cm}^{-1}$) and $2D$ ($2700 \text{ cm}^{-1}$) graphite peaks. Finally, the graphite flake is lifted using AB glue, overcoming the van der Waals adsorption force between the graphite flake and silicon surfaces[14,16]. The lifted graphite flake is then flipped 180 degrees for Raman characterization (LabRAM HR Evolution Raman spectrometer from HORIBA, with resolution of $0.1 \text{ cm}^{-1}$, laser wavelength of 532 nm, grating of 1800 gr/mm, acquisition time of 4 s and spot diameter of 1 μm). The aim is to determine the presence of any observable damage to the slide graphite flake surface based on the presence of a $D$ peak ($1350 \text{ cm}^{-1}$) on the slid graphite flake surface (Fig. 5i).

## The setup of finite element simulation of edge warping mechanism

The FEM simulation model is consistent with the experiments in scale and consist of Au cap, graphite flake and nanostructured silicon substrates, which is shown in Fig. 3a. The Au cap and graphite flake are both $4 \times 4$ μm in size and are bonded together. The thickness of Au cap is 100 nm, and graphite flake's thickness is 200 nm. The nanostructured silicon substrate is $4.8 \text{ μm} \times 4.8 \text{ μm}$ in size and has a thickness of 50 nm without rough peaks. The rough peaks on the nanostructured silicon surface are 7 nm in height and the shape

function for its cross section is $y = A\sin\left(\frac{\pi x}{L}\right)^n$, in which $A = 7$ nm, $L = 350$ nm and $n = 10$. Distance between rough peaks is 400 nm, so there's $12 \times 12$ rough peaks in total and $10 \times 10$ of them are below the graphite mesa consisting of graphite flake and Au cap. The Young's Modulus and Poisson's ratio of Au and Si are 79.5 GPa and 0.42, 190 GPa and 0.278, respectively[30]. The graphite flake is set to be orthotropic with stress-strain relation[31–34]

$$\begin{pmatrix} \sigma_x \\ \sigma_y \\ \sigma_z \\ \tau_{xy} \\ \tau_{yz} \\ \tau_{xz} \end{pmatrix} = \begin{bmatrix} 1060 & 180 & 15 & 0 & 0 & 0 \\ 180 & 1060 & 15 & 0 & 0 & 0 \\ 15 & 15 & 36.5 & 0 & 0 & 0 \\ 0 & 0 & 0 & 440 & 0 & 0 \\ 0 & 0 & 0 & 0 & 4 & 0 \\ 0 & 0 & 0 & 0 & 0 & 4 \end{bmatrix} \begin{pmatrix} \varepsilon_x \\ \varepsilon_y \\ \varepsilon_z \\ \gamma_{xy} \\ \gamma_{yz} \\ \gamma_{xz} \end{pmatrix}, \quad (1)$$

where the elastic modulus is given in units of GPa. The van der Waals interaction between the graphite flake and silicon surface is derived from Lennard–Jones (LJ) potential and simulated with cohesive surface interaction[16], see Supplementary Section 8.1 for details. In the simulation, bottom surface of the substrate is fixed in all directions and the center of the graphite mesa's top surface with diameter of 50 nm, which is in contact with AFM tips, is fixed in the in-plane direction and applied a normal load. During the simulation steps, negligible load is firstly applied to make graphite mesa and the substrate contact, then a normal load of 20 μN is applied gradually. The calculation of finite element method is carried out using finite element software Abaqus/Standard 2020 with implicit solver. Contact between graphite flake and substrate is setup as surface-to-surface contact with small sliding, using direct Lagrangian multiplier method as constraint enforcement method, which enforces a given pressure-overclosure behavior for each constraint, without approximation or use of augmentation iterations. The contact is also set up to be frictionless under structural superlubricity condition. 8-node linear hexahedral solid element with reduced integration is used in the model and the number of elements is 776900, which the average size of the elements is about 20 nm. The mesh convergence analysis of our FEM model in Supplementary Section 8.2 shows that as the mesh is refined, the rate of change of the two characteristic parameters gradually decreases and tends to converge, and sufficient accuracy can be obtained under the number of elements we use.

## The setup of molecular dynamics simulation of single rough peak adhesive force

The MD simulation model consists of the silicon substrate and AB-stacking bilayer graphene. The morphology of the substrate is based on the experimental characterization (Fig. 3e, f, which are characterized by AFM, Cypher S-Oxford Instruments). Since the equilibrium distance of the vdW interaction between silicon and graphite is subnanometer and it decays with distance to the 6 powers, we used a miniature simulation model to ensure the simulation efficiency. The 2-nm-thick substrate used in our simulation is the tip of the realistic rough peak (usually with height 7 nm), and only bilayer graphene is used. The simulation box has the size of 60 nm × 60 nm × 7 nm. There are $4 \times 10^5$ atoms for the simulation system. Periodic boundary conditions are applied to $x$ and $y$ directions. The quasi-static MD simulations are performed using LAMMPS[35]. The interlayer interaction between graphene/graphene and graphene/silicon is described by Lennard–Jones potential[36]. The intralayer interaction of graphene and substrate is described by REBO force field[37] and the modified Tersoff force field[38], respectively. The bottom atomic layer of the substrate is fixed as a rigid body. Springs with spring constant $k_z = 2.7$ N/m are tethered to each carbon atom of the upper graphene layer to mimic the elastic behavior of bulk graphite[39]. To extract the maximum adhesion

between the graphene and silicon substrate, we used the quasi-static simulation protocol[40]. At each step the rigid bottom atomic layer is displaced by $-0.02$Å along $z$ direction, followed by minimizing the total energy of the whole system with FIRE algorithm[41] until the forces acting on each atom reduces below the tolerance $10^{-4}$ eV/Å. In addition, Supplementary Section 9 carries out layer-number dependent MD calculations of adhesion at graphene/silicon interface, the results show that the system using bilayer graphene already gives the saturated value of adhesion.

## Data availability

The data supporting the findings of this study are available within the paper and the Supplementary Information. Source data are provided with this paper.

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

## Acknowledgements

The authors wish to acknowledge the financial support by National Natural Science Foundation of China (11572173, Q.Z., 11890671, Q.Z., 51961145304, Q.Z., 11921002, Q.Z., 11890673, M.M., 11772168, M.M., 12002216, X.J.), the Shenzhen Fundamental Research Key Project (JCYJ20200109150608043, M.M.), and the GuangDong Basic and Applied Basic Research Foundation (2020A1515110995, X.J.).

## Author contributions

Q.Z., M.M., and X.H. designed the experimental aspects of the study. X.H. performed the experiments and analyzed the experimental data. K.X. assisted in the preparation of nanostructured silicon surface. Z.T. assisted in XPS characterization of nanostructured surface, and D.P. assisted in tribological experiments and characterization; T.L. designed and conducted the finite element method simulations of the edge warping mechanism; J.W. conducted the molecular dynamics simulations of adhesion between graphite and silicon nanorough peak; X.X. and B.L. help analyzed the simulation results and mechanism. All authors contributed to the writing of this manuscript.

## Competing interests

The authors declare no competing interests.

## Additional information

**Peer review information** : *Nature Communications* thanks Ernst Meyer and the other, anonymous, reviewer(s) for their contribution to the peer review of this work. A peer review file is available.

