## [Peer Review File · Nature Communications]

Robust microscale structural superlubricity between graphite and nanostructured surfaceREVIEWER COMMENTS

Reviewer #1 (Remarks to the Author):

This manuscript describes an interesting observation from a very carefully designed nano-friction testing. It is known that friction on graphite / graphene basal plane is extremely low when the counter-surface is incommensurate with the hexagonal lattice - as the author mentioned, this is often called structural superlubricity. But friction at the edges of graphite and graphene is quite high. To avoid the edge effect, the authors designed nanostructured substrate, placed a graphite flake, and then pressed the center region which caused the edge region of the flake to lift a little bit (the authors called this "edge warping"); thus the detrimental edge effect was avoided in friction test. What a clever design of the experiment. Then they supported this concept with finite element analysis and molecular dynamics simulations. This reviewer highly noted the amount of work dedicated by students and postdocs to demonstrate this idea, but, it is difficult to find significant scientific meaning or new insights attained by this work. In the introduction, the authors claimed a revolutionary approach is needed to solve lubrication challenges in MEMS, microsensors, microrobots, etc; but, it is difficult to see how this method could be generalized to such devices that may not use the graphite flakes.

Reviewer #2 (Remarks to the Author):

First of all, it is great to see this group's steady progress and continuous contributions to the community of tribology, which makes us better understanding of structural superlubricity, a revolutionary solution of friction and wear problems. Based partially on previous studies, this work goes further and presents a very interesting experimental study combined with FEM/MD computational modeling showing the following important findings: (1) a robust SSL state between microscale graphite flakes and nanostructured silicon surfaces without observable wear can be achieved under ambient condition. (2) Through FEM and MD simulations, the authors believe that it is attributed to the edge warping of graphite flake on the nanostructured surface under concentrated force, which eliminates the edge interaction between the graphite flake and the substrate. (3) Such a surface modification method could be general, which promotes the general application of SSL technology. The work seems to be done very carefully and consistently and the paper is well written. However, some essential information is still missing. Based on the overall quality, it is my judgment that this paper should be published in nature communication after some major revisions

Major points

1. In FEM modeling, the contact, sliding and frictional simulations are not an easy task. Definitely, this project is a large sliding problem and some key pieces are missing. Which method is used to solve the contact problem, penalty method or the augmented Lagrangian method? If the latter, a segregated or coupled solution method is used? How to generate the mesh, especially in the interface (For instance, authors can refer to the figure 2 in PHYSICAL REVIEW E 70, 026117 (2004) or Journal of the Mechanics and Physics of Solids 53 (2005) 2385-2409)? For this type of large sliding problem, how do authors guarantee that the results are converged without providing this detailed information? Also, which software did authors use, in-house code or commercial software (which version)?
2. In MD simulations, AB-stacking bilayer graphene is adopted instead of graphite flakes, which is not consistent with experiment. Also, there might be thickness-dependent behavior showing up here. The authors did not discuss and exclude this effect.
3. In FEM modeling, the concentrated normal load of 20 μN is applied to the center of the graphite flake, which can be approximated to a circle with radius 0.124 μm . However, the loading conditions in MD simulations are confusing. As mentioned in "In order to obtain the de-adhesive force generated by the bending moment of the graphite flake on the rough peak of nanostructured silicon surface under the concentrated force, we modified the boundary conditions based on the finite element method (FEM) simulation setup in Fig. 3a. By constraining the relative displacement in the z direction between the bottom surface of the graphite flake", so in each MD run, the z distance between the bottom surface of the graphite flake is fixed. However, it looks misleading that in figure 3g and h, the concentrated normal load is applied to the center of the bilayer

graphene.

Other minor points:

1. I am not experimental expert. So I am confused that in figure 1e we should observed the surface topography of nanostructured surface with a regular pattern corresponding to 32 rough peaks.
2. References should be consistent. The titles of some references are in capital letters.

Reviewer #3 (Remarks to the Author):

The describes friction measurements of graphite flakes on nanostructured silicon surfaces. The authors find robust superlubricity conditions with friction coefficients 10^{-4} and friction values below 1micronNewton. In contrast, they find increased friction on flat silicon surfaces. Comparison with finite element calculations indicate that warping occurs on the nanostructured surfaces, which seems to reduce the remaining friction forces at the border of the contacts. On flat silicon, the edge forces are dominating, which gives rise to the increased friction. In summmary, this is a surprising observation, which might indicate a way to use superlubricity in applications across larger contacts. Therefore, I recomment to publish this paper with major revisions.

Points to be adressed:

The fabrication of the nanostructured surfaces involves some solvents and polystyrene spheres. Can the authors exclude transfer of polystyrene to the silicon surface? If parts of PS would be on the surface, this might lead to different tribological conditions. Surface analysis by XPS might clarify this point.

-It might be also of interest to change the sizes of the PS spheres and change the separations of the nanoasperities. How does this influence the superlubricity conditions?

-Measurements were performed in humid conditions. Capillary condensation might play a role. How thick were the water films on the silicon surfaces and how is water excluded in the contact zone?

Below we respond (in Black) all points (in Italic) raised by the referees:

Reviewer #1

Comment: “This manuscript describes an interesting observation from a very carefully designed nano-friction testing. it is known that friction on graphite / graphene basal plane is extremely low when the counter-surface is incommensurate with the hexagonal lattice - as the author mentioned, this is often called structural superlubricity. But friction at the edges of graphite and graphene is quite high. to avoid the edge effect, the authors designed nanostructured substrate, placed a graphite flake, and then pressed the center region which caused the edge region of the flake to lift a little bit (the authors called this "edge warping"); thus the detrimental edge effect was avoided in friction test. What a clever design of the experiment. Then they supported this concept with finite element analysis and molecular dynamics simulations. This reviewer highly noted the amount of work dedicated by students and postdocs to demonstrate this idea,”

Response: We would like to thank Reviewer 1 for his/her careful reading and positive comments.

“but, it is difficult to find significant scientific meaning or new insights attained by this work. In the introduction, the authors claimed a revolutionary approach is needed to solve lubrication challenges in MEMS, microsensors, microrobots, etc; but, it is difficult to see how this method could be generalized to such devices that may not use the graphite flakes.”

Response: We thank the reviewer for the question which makes us realized that we didn't make the novelty of our findings clear enough. Here we would like to explain these points in more details. In traditional tribology and structural superlubricity theories, it is generally believed that rougher surfaces lead to higher friction and lead to wear. However, this study found that the graphite flake/flat

silicon interface has a tendency to exhibit high friction ($>5 \mu\text{N}$) and wear. On the other hand, the microscale graphite flake/nanostructured silicon interface consistently displays a robust structural superlubric (SSL) state under ambient conditions, characterized by friction force less than $1 \mu\text{N}$, a friction coefficient of approximately 10^{-4} , and no observable wear. This work not only challenges traditional tribology and structural superlubricity theories which usually predicts a larger friction for rougher surface, but also reduces the roughness requirements for SSL technology which used be a challenge for practical application of SSL, but also demonstrate that a graphite flake with a single crystal surface that does not come into edge contact with the substrate can consistently achieve robust SSL state with any non-van der Waals material in atmospheric conditions.

Furthermore, the study provides a general surface modification method to achieve robust SSL state between graphene flakes and non-vdW materials, further promoting the widespread application of SSL technology. For example, as an application of SSL, superlubric generator (SLG)^{1,2} is designed to produce stable high current density and long-lifespan output under relative sliding through SSL technology, and the conversion efficiency is close to 100%^{1,2}. Recently, the Schottky superlubric generator (S-SLG), that is, the sliding contact between micro-sized graphite flakes and n-type silicon in the SSL state, was proposed as a physical prototype of SLGs (Supplementary Fig. 20a)². However, the edges of graphite flake will cause high friction and wear with a certain probability, leading to the failure of the SLG, as shown in Supplementary Fig. 20a. Here, we demonstrate our surface modification method to optimize the original SLG to avoid high friction and wear, as shown in Supplementary Fig. 20b. To promote SLG to large-scale batch applications, we can connect the centers of multiple graphite flakes placed on nanostructured n-Si with columnar Ni through sacrificial layer and electroplating methods to realize the center loading (inducing warpage curved edges) and integrated connections of surface-modified SLGs, as shown in Supplementary Fig. 20c. While the microfabrication process of such structure is under operation and takes time, we believe that with the mechanism

revealed in our present manuscript, it is reasonable to believe that the friction and wear will greatly be reduced with the help of edge warping.

Supplementary Fig. 20 Optimized superlubric generators (SLGs) through surface modification. **a** Schematic diagram of original SLG. **b** Schematic diagram of surface-modified SLG, nanostructures are prepared on the n-Si surface. **c** Assembly of surface-modified SLGs, the centers of multiple graphite flakes placed on nanostructured n-Si are connected with columnar nickel by sacrificial layer and electroplating methods.

A description of the above scientific significance and application example was added in the Conclusion and Abstract sections of the revised manuscript and in Supplementary Section 11.

Reviewer #2

Comment: “First of all, it is great to see this group's steady progress and continuous contributions to the community of tribology, which makes us better understanding of structural superlubricity, a revolutionary solution of friction and wear problems. Based partially on previous studies, this work goes further and presents a very interesting experimental study combined with FEM/MD computational modeling showing the following important findings: (1) a robust SSL state between microscale graphite flakes and nanostructured silicon surfaces without observable wear can be achieved under ambient condition. (2) Through FEM and MD simulations, the authors believe that it is attributed to the edge warping of graphite flake on the nanostructured surface under concentrated force, which eliminates the edge interaction between the graphite flake and the substrate. (3) Such a surface modification method could be general, which promotes the general application of SSL technology. The work seems to be done very carefully and consistently and the paper is well written. However, some essential information is still missing. Based on the overall quality, it is my judgment that this paper should be published in nature communication after some major revisions”

Response: We would like to thank Reviewer 2 for his/her careful reading and positive comments.

“Major points:

1. In FEM modeling, the contact, sliding and frictional simulations are not an easy task. Definitely, this project is a large sliding problem and some key pieces are missing. Which method is used to solve the contact problem, penalty method or the augmented Lagrangian method? If the latter, a segregated or coupled solution method is used? How to generate the mesh, especially in the interface (For instance, authors can refer to the figure 2 in PHYSICAL REVIEW E 70, 026117 (2004) or Journal of the Mechanics and Physics of Solids 53 (2005) 2385–2409)? For this type of large sliding problem, how do authors guarantee that the results are converged without providing this detailed

information? Also, which software did authors use, in-house code or commercial software (which version)?”

Response: We thank the reviewer for acknowledge the effort we have spent in the FEM simulation. As suggested, we have supplemented the setting of key pieces of our FEM model in detail in the Method section of the revised manuscript. In response to the specific questions listed, our one-by-one replies are as follows.

(i) The purpose of our FEM simulation is to qualitatively prove the mechanism that the edge of the graphite flake warps rather than adhere to the substrate under central loading and using the mechanism to explain phenomena observed in the experiment. Therefore, our FEM model is a static model that only focuses on the contact problem and doesn't contain the process of graphite flakes sliding on the substrate.

(ii) Contact between graphite flake and substrate is set up as surface-to-surface contact with small sliding, using direct Lagrangian multiplier method as constraint enforcement method, which enforces a given pressure-overclosure behavior for each constraint, without approximation or use of augmentation iterations.

(iii) We thank the reviewer for having provided us these two nice works which have added in our revised work. The recommended articles suit well for the meshing of rough surface while keeping the real morphology of the surface. However, for a clearer and more concise analysis of the mechanism, we simplified the nanostructured surface as a plain surface with array of regular and gentle undulations, which is not a real rough surface. The spacing and height of the rough peak array are 400 nm and 7 nm respectively, and the surface shape of each roughness peak is obtained by rotating and sweeping the function $y = A \sin\left(\frac{\pi x}{L}\right)^n$ around the symmetry axis, where $A = 7$ nm, $L = 350$ nm and $n = 10$. The above parameters are statistically obtained from actual measured topography from Fig. 2b (see Supplementary Section 4 for details). We have added these

explanations to the revised manuscript. The mesh is generated using sweep method, and cohesive behavior is used to simulate the van der Waals interaction. As a supplement, MD simulations that considering the real topography around the contact region on the apex of the nanostructure were performed to describe the de-adhesion process at the atomic scale. In addition, the finite element simulation for the eccentric loading case in Fig. 4 yields that the minimum warpage height Δh_{\min} drops rapidly to negative when $\delta \geq 1.2 \mu\text{m}$, which explains the failure of the SSL state when $\delta = 1.2 \mu\text{m}$ in the experiment (Fig. 4f) and justifies our simplification of the model.

(iv) Frictionless contact also be set up under structural superlubricity condition, making the simulation not too complicated to converge. 8-node linear hexahedral solid element with reduced integration is used in the model and the number of elements is 776900, which the average size of the elements is about 20nm. In order to analyze the mesh convergence of our model, we conducted simulations with different number of elements under the same setting, and extracted two critical parameters in the simulation results, namely the maximum stress $U_{\max}^{(\text{Mises})}$ on the bottom surface of graphite flake (upper) and the minimum displacement in height Δh_{\min} along the graphite flake edge (lower), as shown in the Supplementary Fig. 14. The results show that with the refinement of the mesh, the rate of change of the two characteristic parameters gradually decreases and tends to converge, and sufficient accuracy can be obtained under the number of elements we use.

Supplementary Fig. 14 Mesh convergence analysis of our FEM model. The maximum Mises stress $U_{\max}^{(\text{Mises})}$ on the bottom surface of the graphite flake (upper) and the minimum displacement in height Δh_{\min} along the graphite flake edge (lower) for different element numbers.

(v) We used the commercial software Abaqus/Standard version 2020 for calculation.

“2. In MD simulations, AB-stacking bilayer graphene is adopted instead of graphite flakes, which is not consistent with experiment. Also, there might be thickness-dependent behavior showing up here. The authors did not discuss and exclude this effect.”

Response: The van der Waals interaction between the graphite and the substrate is a short-ranged interaction. In our MD simulations, it is represented by Lennard-Jones potential, $V_{ij} = 4\varepsilon \left[\left(\frac{\sigma}{r_{ij}} \right)^{12} - \left(\frac{\sigma}{r_{ij}} \right)^6 \right]$, where ε is the energy well depth, and σ is the characteristic length. Typically, the value of σ is 3 \AA –

comparable to the interlayer distance of graphite. According to the formula, V_{ij} becomes the high order small quantity ($< 10\% \epsilon$) for $r_{ij} \gtrsim 2\sigma$; for $r_{ij} = 3\sigma$, the interaction becomes negligibly small, $V_{ij} \approx 0.5\% \epsilon$. Thus, the use of bilayer graphene is believed to be accurate in estimating the adhesion.

To validate the discussion above, we performed additional simulations with 1 and 4 layers of graphene as shown in Supplementary Fig. 18. It can be seen from the simulation results that the system using bilayer graphene already gives the saturated value of adhesion.

Supplementary Fig. 18 Molecular dynamics (MD) simulations of adhesion at graphene/silicon interfaces with different numbers of layers.

The above results and analyses are added in the Supplementary Section 9 and mentioned in Method section of revised manuscript.

“3. In FEM modeling, the concentrated normal load of $20 \mu\text{N}$ is applied to the center of the graphite flake, which can be approximated to a circle with radius $0.124 \mu\text{m}$. However, the loading conditions in MD simulations are confusing. As mentioned in “In order to obtain the de-adhesive force generated by the bending moment of the graphite flake on the rough peak of nanostructured silicon surface under the concentrated force, we modified the boundary conditions based on the finite element method (FEM) simulation setup in Fig. 3a. By constraining the relative displacement in the z direction

between the bottom surface of the graphite flake”, so in each MD run, the z distance between the bottom surface of the graphite flake is fixed. However, it looks misleading that in figure 3g and h, the concentrated normal load is applied to the center of the bilayer graphene.”

Response: We thank the reviewer for pointing out the misleading description in our previous manuscript. The molecular dynamics (MD) model in Fig. 3g is mainly to calculate the maximum adhesion force between a local rough peak and the bottom surface of the graphite flake. The MD simulation model was established according to the morphology of the rough peak in experiments (Figs. 3e and f). The simulated adhesive force versus distance respect to the equilibrium interlayer spacing is shown in Fig. 3h (blue solid line), where the maximum adhesion is 46 nN (blue dotted line). No concentrated force was applied during the MD simulation. The misleading original black arrows in Fig. 3g have been removed in the revised manuscript.

To quantify the de-adhesive force created by the bending moment of the graphite flake on the rough peak of a nanostructured silicon surface under concentrated force, we modified the boundary conditions of the finite element method (FEM) simulation depicted in Fig. 3a. By constraining the relative displacement in the z-direction between the bottom surface of the graphite flake and the rough peak of the nanostructured silicon, we calculated the Mises stress distribution across the diagonal section under central normal loading, as shown in the inset of Fig. 3h. Our simulation results revealed that along the diagonal direction, the two closest rough peaks to the loading center experience pressure, while the third rough peak generates the balancing reaction force (i.e., the de-adhesive force), which was found to be approximately 178 nN (represented by the red dotted line in Fig. 3h), surpassing the maximum adhesion force of 46 nN. As a result, de-adhesion occurs at the third rough peak position. Moreover, the third rough peak closest to the loading center experiences the lowest de-adhesive force, and subsequent de-adhesion occurs at all peripheral rough peaks after de-

adhesion occurs at the third rough peak. These MD and FEM results support the de-adhesion process at the atomic scale.

A clearer explanation of the above for MD and FEM simulations of Figs. 3g and h has been added to the revised manuscript.

“Other minor points:

1. I am not experimental expert. So I am confused that in figure 1e we should observed the surface topography of nanostructured surface with a regular pattern corresponding to 32 rough peaks.”

Response: In Fig. 1e, the nanostructures are randomly distributed. This is a result of our serendipitous observation that the nanostructured silicon surface exhibits lower friction with graphite flake compared to the flat silicon surface. To gain further insights into this phenomenon and to explore the underlying mechanism, we employed polystyrene microspheres as a mask to fabricate nanostructures with uniform peak heights and distribution on the silicon surface (the fabrication process is detailed in the Supplementary Section 2 and Methods section). Thus, our subsequent analyses and tests depicted in Figs. 2-5 are based on the nanostructured surface prepared as described above, whereas Fig. 1e is not. To clarify this distinction, we have added further emphasis in the revised manuscript.

“2. References should be consistent. The titles of some references are in capital letters.”

Response: We thank the reviewe for point out this mistake. The format of all references has been revised accordingly.

Reviewer #3

Comment: “The describes friction measurements of graphite flakes on nanostructured silicon surfaces. The authors find robust superlubricity conditions with friction coefficients 10^{-4} and friction values below 1micronNewton. In contrast, they find increased friction on flat silicon surfaces. Comparison with finite element calculations indicate that warping occurs on the nanostructured surfaces, which seems to reduce the remaining friction forces at the border of the contacts. On flat silicon, the edge forces are dominating, which gives rise to the increased friction. In summmary, this is a surprising observation, which might indicate a way to use superlubricity in applications across larger contacts. Therefore, I recommend to publish this paper with major revisions.”

Response: We would like to thank Reviewer 3 for his/her careful reading and positive comments.

“Points to be adressed:

The fabrication of the nanostructured surfaces involves some solvents and polystyrene spheres. Can the authors exclude transfer of polystyrene to the silicon surface? If parts of PS would be on the surface, this might lead to different tribological conditions. Surface analysis by XPS might clarify this point.”

Response: We thank the reviewer for the suggestion and we have performed the surface analysis accordingly. After the fabrication process of nanostructured silicon surface, we use acetone, alcohol and deionized water ultrasonic cleaning to remove the residual solvents and polystyrene spheres on the nanostructures, and the AFM morphology of the fabricated nanostructured after final ultrasonic cleaning is shown in Supplementary Fig. 2g. The nanostructures are uniformly distributed, and the peak height is uniform. To demonstrate that solvents and polystyrene spheres were completely expelled after the fabrication of the

nanostructured surface and exclude the transfer of polystyrene to the silicon surface resulting in different tribological conditions, we used a surface elemental analysis method with a shallower detection depth, that is, X-ray photoelectron spectroscopy (XPS, Thermo Fisher, Model: ESCALAB 250Xi, Source gun type: Al K Alpha, Spot size: 500 μm , Lens mode: Standard, Pass energy 30.0 eV) to characterize the prepared nanostructured silicon surface, as shown in Supplementary Fig. 3. First of all, according to the narrow scan spectra results of C1s in Supplementary Fig. 3b we measured, the peak energy is 284.82 eV, which corresponds to the adventitious hydrocarbon contamination (C aromatic and aliphatic)³. Compared with the standard C1s XPS spectrum features of polystyrene (Supplementary Table 1)⁴, we found that the peak at 291.27 eV corresponding to the shake-up transition does not appear in our measurement results^{4,5}, which indicates that no polystyrene remains on the nanostructured silicon surface.

Supplementary Fig. 3 X-ray photoelectron spectroscopy (XPS) characterization of prepared nanostructures on silicon surface. **a** Full-scan spectra (Energy step size: 1.000 eV). **b-d** are the narrow scan spectra (Energy step size: 0.050 eV) for the three characteristic peaks of C1s, O1s and Si2p respectively.

Supplementary Table 1 Standard spectral features of polystyrene⁴.

Element/ Transition	Peak Energy (eV)	Peak Width FWHM (eV)	Peak Area (eV×Cts/s)	Sensitivity Factor	Concentration (at. %)	Peak Assignment
C 1s	284.72	1.31	285262	1	71.4	C aromatic
C 1s	285.00	1.59	87143	1	21.8	C aliphatic
C 1s	291.27	1.13	27011	1	6.8	shake-up transition

The above results and analyses are added in the Supplementary Section 2 and mentioned in Method section of revised manuscript.

“It might be also of interest to change the sizes of the PS spheres and change the separations of the nanoasperities. How does this influence the superlubricity conditions?”

Response: We thank the reviewer for the interesting suggestion. We first performed finite element method (FEM) simulations of the warping of graphite flakes that placed on nanostructured silicon with different separation under central loading. The distance between nanoasperities gradually increases from 400nm in the experiment to 800nm with the same shape and size. Considering the cost, compared with the simulation work in the main text, this part of the simulation uses a coarser mesh division and model. The simulated results as shown in Supplementary Fig. 17. From the results we can conclude that the warpage height of the graphite flake edge decreases with the increase of the separation between the nanorough peaks (Supplementary Figs. 17c-e). When the nanorough peaks becomes sparse enough ($d > 600$ nm), even increasing the loading at the center of the graphite flake cannot warp the entire edge above the horizontal plane (Supplementary Figs. 17c and e). As a result, the part of the edge that sinks below the horizontal plane will interact with the nanorough peaks of the substrate (collisions, bonding and breaking, etc.), regenerating high friction and wear, thereby disrupting the robust SSL state. Therefore, controlling the size and separation of the nano-asperities is of great significance for the surface modification method in this work to achieve robust SSL.

However, the central contribution of this work is the proposal and demonstration of a novel phenomenon, the edge warping of graphite flakes on nanostructured surfaces under concentrated force, which effectively eliminates the high friction and wear that arises from strong edge-substrate interactions. Therefore, using polystyrene microspheres of different sizes to adjust the

separation of nanostructures, studying its effect on superlubricity conditions, and verifying the laws predicted by the above finite element simulations are the work that needs to be carried out in the next stage.

Supplementary Fig. 17 Simulation of graphite flakes under center loading on nanostructures with different separations. **a** Schematic diagram of the finite element simulation model, except that the separation of the nanostructure (that is, the distance d between rough peaks) is set to 400 nm 500 nm, 600 nm, 700 nm and 800 nm respectively, the other settings are consistent with Fig. 3 in the main text. Several feature locations are marked with white dots in the figure, which are Center (center of the bottom surface), Middle (center of the edge), and Corner (corner of the

edge). The feature lines connecting the white dots are diagonal of bottom surface (blue dotted line) and edge (yellow solid line). **b** and **c** show the displacement in height distribution along the bottom interface of graphite flake across diagonal section (along the blue dotted line in **a**) and edge section (along the yellow solid line in **a**) under different separations of the nanostructures with a normal load of 20 μN , respectively. **d** and **e** represent the corner warpage height Δh_{corner} and center warpage height Δh_{middle} of the graphite flake edge under different normal forces, respectively.

The above results and analyses are added in the Supplementary Section 8.5 and mentioned in revised manuscript.

“Measurements were performed in humid conditions. Capillary condensation might play a role. How thick were the water films on the silicon surfaces and how is water excluded in the contact zone?”

Response: We thank the reviewer for this interesting question. The contacts used in our experiments are composed of graphite and silicon surface. When exposed to ambient condition, e.g., the temperature of 25 ± 1 °C and relative humidity of $25 \pm 1\%$ as used in our experiment, the graphite will be covered with a water film of which the thickness is less than 1 nm.⁶ For the silicon surfaces, there have also been detailed experimental measurements which shows that the thickness of the water film is also less than 1 nm⁷⁻¹¹. To illustrate this point, we used a commercial instrument (DataPhysics-OCA 25) to measure the contact angle of water on the surface of the prepared nanostructured silicon under the same experimental environment. The test was done 4 times (at different positions), where the optical microscope observation during the test is shown in Supplementary Fig. 19. The measured contact angle and experimental parameters are shown in Supplementary Table 3, which presents a weak hydrophobicity (CA~ 92.30°-101.48°). Therefore, such thickness, e.g., less than 2 nm in total, is much less than the typical height of 7 nm for the nanostructures used in our experiments, so that

the water film does not completely submerge the nanostructures, and the water film in the bottom region of the nanostructure surface does not directly contact the bottom surface of graphite flakes. In addition, even there was a thin water film covering the apex of the nanostructures, under a normal pressure of few hundreds MPa as used in our frictional test, the water film would be extruded from the contact between the graphite and the apex of the nanostructures, as it can sustain a normal load less than 20 MPa when confined between a nanoscale tip/pillar and graphite¹²⁻¹⁵. As a consequence, capillary condensation has no effects in the friction between the nanostructured silicon surface and graphite. For that between flat silicon surface and graphite, it may play a role, which however is beyond the scope of our present study. Part of the above analysis was added in the revised manuscript.

Supplementary Fig. 19 Contact angle measurements of water on prepared nanostructured silicon surface. a-d are the optical observations during the tests at different positions.

Supplementary Table 3 Measured contact angle and experimental parameters of water on prepared nanostructured silicon surface (temperature of 25 ± 1 °C and

relative humidity of 25± 1%).

Experiments No.	CA(L)[°]	CA(R)[°]	CA(M)[°]	Drop volume [μL]	Contact diameter [mm]	Drop height[mm]
1	95.66	95.64	95.67	1.58	1.75	0.94
2	100.69	100.92	100.46	1.86	1.77	1.03
3	101.48	101.41	101.55	1.86	1.75	1.04
4	92.30	92.02	92.58	1.77	1.86	0.95

The above analyses are added in the Supplementary Section 10 and mentioned in revised manuscript.

References

- 1 Huang, X., Lin, L. & Zheng, Q. Theoretical study of superlubric nanogenerators with superb performances. *Nano Energy*, 104494 (2020).
- 2 Huang, X. *et al.* Microscale Schottky superlubric generator with high direct-current density and ultralong life. *Nature Communications* **12**, doi:10.1038/s41467-021-22371-1 (2021).
- 3 Jensen, D. S. *et al.* Silicon (100)/SiO₂ by XPS. *Surface Science Spectra* **20**, 36-42 (2013).
- 4 Girardeaux, C. & Pireaux, J. J. Analysis of polystyrene (PS) by XPS. *Surface Science Spectra* **4**, 130-133, doi:10.1116/1.1247812 (1996).
- 5 Bhatia, Q. S., Pan, D. H. & Koberstein, J. T. Preferential surface adsorption in miscible blends of polystyrene and poly (vinyl methyl ether). *Macromolecules* **21**, 2166-2175 (1988).
- 6 Wang, K. *et al.* Structural superlubricity with a contaminant-rich interface. *Journal of the Mechanics and Physics of Solids* **169**, doi:10.1016/j.jmps.2022.105063 (2022).
- 7 Chen, L., He, X., Liu, H., Qian, L. & Kim, S. H. Water adsorption on hydrophilic and hydrophobic surfaces of silicon. *The Journal of Physical Chemistry C* **122**, 11385-11391 (2018).
- 8 Chen, L. *et al.* Dependence of water adsorption on the surface structure of silicon wafers aged under different environmental conditions. *Physical Chemistry Chemical Physics* **21**, 26041-26048 (2019).
- 9 Cao, P., Xu, K., Varghese, J. O. & Heath, J. R. The microscopic structure of adsorbed water on hydrophobic surfaces under ambient conditions. *Nano letters* **11**, 5581-5586 (2011).
- 10 Chen, L., He, X., Liu, H., Qian, L. & Kim, S. H. Water Adsorption on Hydrophilic and Hydrophobic Surfaces of Silicon. *J. Phys. Chem. C* **122**,

- 11385-11391, doi:10.1021/acs.jpcc.8b01821 (2018).
- 11 Cao, P. G., Xu, K., Varghese, J. O. & Heath, J. R. The Microscopic Structure of Adsorbed Water on Hydrophobic Surfaces under Ambient Conditions. *Nano Lett.* **11**, 5581-5586, doi:10.1021/nl2036639 (2011).
- 12 Li, J., Cao, W., Li, J. & Ma, M. Fluorination to enhance superlubricity performance between self-assembled monolayer and graphite in water. *Journal of Colloid and Interface Science* **596**, 44-53 (2021).
- 13 Li, J., Cao, W., Li, J., Ma, M. & Luo, J. Molecular origin of superlubricity between graphene and a highly hydrophobic surface in water. *The journal of physical chemistry letters* **10**, 2978-2984 (2019).
- 14 Liu, B. *et al.* Negative friction coefficient in microscale graphite/mica layered heterojunctions. *Science Advances* **6**, eaaz6787 (2020).
- 15 Li, J., Cao, W., Li, J., Ma, M. & Luo, J. Molecular Origin of Superlubricity between Graphene and a Highly Hydrophobic Surface in Water. *J. Phys. Chem. Lett.* **10**, 2978-2984, doi:10.1021/acs.jpcclett.9b00952 (2019).

REVIEWERS' COMMENTS

Reviewer #2 (Remarks to the Author):

The authors have addressed all my concerns in details, which meet my standard and expectations. So this manuscript is good to go for publication.

Reviewer #3 (Remarks to the Author):

The authors have addressed all the points raised by the referees. In particular, they show good evidence by XPS data that the processing of the samples leads to clean graphite surfaces (see suppl. Fig. 3). They investigate the influence of humidity and water layers (See suppl fig. 19). They also present the dependence on the distance between nanoasperities and its influence on superlubricity (See supp. Fig. 17). Therefore, I recommend to publish the paper in its present form.

Below we respond (in Black) all points (in Italic) raised by the referees:

Reviewer #2

Comment: “The authors have addressed all my concerns in details, which meet my standard and expectations. So this manuscript is good to go for publication.”

Response: We would like to thank again to Reviewer 2 for his/her careful reading, valuable revision suggestions and positive comments.

Reviewer #3

Comment: “The authors have addressed all the points raised by the referees. In particular, they show good evidence by XPS data that the processing of the samples leads to clean graphite surfaces (see suppl. Fig. 3). They investigate the influence of humidity and water layers (See suppl fig. 19). They also present the dependence on the distance between nanoasperities and its influence on superlubricity (See supp. Fig. 17). Therefore, I recommend to publish the paper in its present form.”

Response: We would like to thank again to Reviewer 3 for his/her careful reading, valuable revision suggestions and positive comments.